# AIDE: An Automatic Data Engine for Object Detection in Autonomous Driving

## Abstract

001     *Autonomous vehicle (AV) systems rely on robust percep-*
002 *tion models as a cornerstone of safety assurance. However,*
003 *objects encountered on the road exhibit a long-tailed distri-*
004 *bution, with rare or unseen categories posing challenges to*
005 *a deployed perception model. This necessitates an expen-*
006 *sive process of continuously curating and annotating data*
007 *with significant human effort. We propose to leverage recent*
008 *advances in vision-language and large language models to*
009 *design an Automatic Data Engine (AIDE) that automati-*
010 *cally identifies issues, efficiently curates data, improves the*
011 *model through auto-labeling, and verifies the model through*
012 *generation of diverse scenarios. This process operates it-*
013 *eratively, allowing for continuous self-improvement of the*
014 *model. We further establish a benchmark for open-world*
015 *detection on AV datasets to comprehensively evaluate vari-*
016 *ous learning paradigms, demonstrating our method's supe-*
017 *rior performance at a reduced cost.*

## 1. Introduction

019 Autonomous vehicles (AVs) operate in an ever-changing
020 world, encountering diverse objects and scenarios in a long-
021 tailed distribution. This open-world nature poses a signifi-
022 cant challenge for AV systems since it is a safety-critical
023 application where reliable and well-trained models must be
024 deployed. The need for continuous model improvement
025 becomes apparent as the environment evolves, demand-
026 ing adaptability to handle unexpected events. Despite the
027 wealth of data collected on the road every minute, its effec-
028 tive utilization remains low due to challenges in discerning
029 which data to leverage. While solutions exist for this in in-
030 dustry [1, 2], they are often trade secrets and presumably
031 require significant human effort. Hence, developing a com-
032 prehensive automated data engine can lower entry barriers
033 for the AV industry.

034     Designing automated data engines can be challenging,
035 but the existence of Vision-Language Models (VLMs) and
036 Large Language Models (LLMs) allows new avenues to
037 these hard problems. A traditional data engine can be bro-

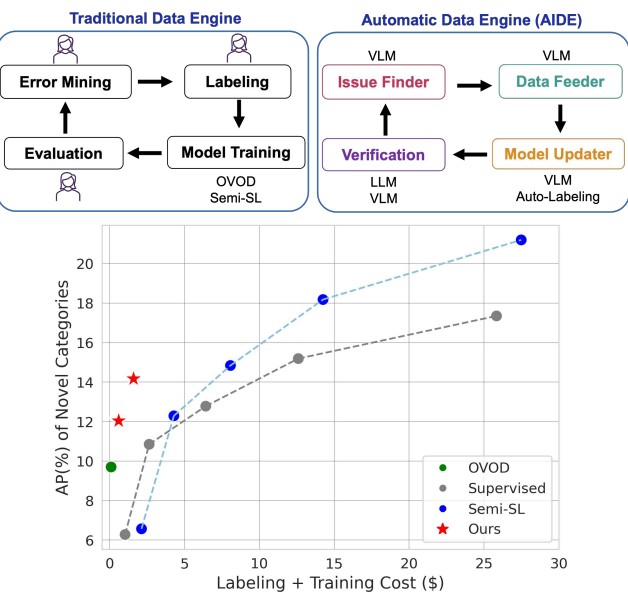

Figure 1. **Top:** Components for DevOp systems for autonomous driving. **Bottom:** With our automatic data system, we can achieve similar performance with less labeling and training costs.

038 ken down into finding issues, curating and labeling data,
039 model training, and evaluation, all of which can benefit
040 from automation. In this paper, we propose an Automati-
041 cally Improving Data Engine (called AIDE) that leverages
042 VLMs and LLMs to automate the data engine. Specifi-
043 cally, we use VLMs to identify the issue, query relevant
044 data, auto-label data, and verify together with LLMs. The
045 high-level steps are shown in Fig. 1 top.

046     In contrast to traditional data engines that rely heavily
047 on extensive human labeling and intervention, AIDE auto-
048 mated the process by utilizing pre-trained VLMs and LLMs.
049 Different from other confidential solutions in industry [1,
050 2], we provide our efficient solutions to lower the entry
051 barrier. While open-vocabulary object detection (OVOD)
052 methods [3, 4] do not require any human annotations, they
053 are a good starting point for detecting novel objects but their
054 performances fall short on AV datasets compared to super-

vised methods. Another line of research on minimizing labeling costs is semi-supervised learning [5, 6] and active learning [7–10]. Although they generate pseudo-labels, the vast amount of data collected on the road is still not fully utilized, in contrast with our method which leverages pretrained VLMs and LLMs for better data utilization.

The detailed steps of AIDE are shown in Fig. 2. In the Issue Finder, we use a dense captioning model to describe the image in detail, then match if the objects in the description are included in the label spaces or the predictions. This is based on the reasonable but previously unexploited assumption that large image captioning models are more robust starting points in zero-shot settings than OVOD (Tab. 3). The next step is to find relevant images that could contain the novel category using our Data Feeder. We find that VLM gives more accurate image retrieval than using image similarity to retrieve images (Tab. 4). We then use our existing label space plus the novel category to prompt the OVOD method, i.e., OWL-v2 [11], to generate predictions on the queried images. To filter these pseudo predictions, we use CLIP to perform zero-shot classification on the pseudo-boxes to generate pseudo-labels for the novel categories. Last, we exploit the LLM, e.g., ChatGPT [12], in Verification to generate diverse scene descriptions given the novel objects. Given the generated description, we again use VLM to query relevant images to evaluate the updated model. To ensure the correctness, we ask humans to review if the predictions of the novel categories are correct. If it is not, we ask humans to provide ground-truth labels, which are used to further improve the model. (Fig. 6)

To verify the effectiveness of our AIDE, we propose a new benchmark on existing AV datasets to comprehensively compare our AIDE with other paradigms. With our Issue Finder, Data Feeder, and Model Updater, we bring 2.3% Average Precision (AP) improvement on the novel categories compared with OWL-v2 without any human annotations and also surpass OWL-v2 by 8.9% AP on known categories (Tab. 1). We also show that with a single round of Verification, our automatic data engine can further bring 2.2% AP on novel categories without forgetting the known categories, as shown in Fig. 1. To summarize, our contributions are two-fold:

- We propose a novel design paradigm for an automatic data engine for autonomous driving as automatic data querying and labeling with VLM and continual learning with pseudo labels. When scaling up for novel categories, this approach achieves an excellent trade-off between detection performance and data cost.
- We introduce a new benchmark to evaluate such automated data engines for AV perception that allows combined insights across multiple paradigms of open vocabulary detection, semi-supervised, and continual learning.

## 2. Related Works

**Data Engine for Autonomous Vehicles (AV)** Exploiting large-scale data collected by AV is crucial to speed up the iterative development of the AV system [13]. Existing literature mostly focuses on developing general [14, 15] learning engines or specific [16] data engines, and most of them [17, 18] mainly focus on the model training part. However, a fully functional AV data engine requires issue identification, data curation, model retraining, verification, etc. A thorough examination reveals a lack of systematic research papers or literature that delves deeply into AV data engines in academia, where a recent survey [13] also underscores the lack of study in this context. On the other hand, existing solutions [1, 2] for AV data systems mainly rely on the design of data infrastructure and still need lots amount of human effort and intervention, thus limiting their maintenance simplicity, affordability, and scalability. In contrast, the present paper exploits the burgeoning progress of vision language models (VLMs) [19–21] to design our data engine, where their strong open-world perception capability largely improves our engine's extendability and makes it more affordable to scale up our AVs on detecting novel categories. To our best knowledge, this paper is also the first work that provides a systematic design of data engines for AVs with the integration of VLMs.

**Novel Object Detection** Conventional 2D object detection has made enormous progress [22, 23] in the last decades, while its closed-set label space makes unseen category detection infeasible. On the other hand, open-vocabulary object detection (OVOD) [4, 24–39] methods promises to detect anything by a simple text prompting. However, their performances are still inferior to closed-set object detection since they must balance the specificity of pretrained categories and the generalizability of unseen categories. To scale up the capacity of open-vocabulary detector (OVD), recent works either pre-train OVD with weak annotations (e.g., image captions) [40], or perform self-training on daily object datasets [41, 42] or web-scale datasets [4, 43]. However, balancing the trade-off between improving the novel categories while mitigating the catastrophic forgetting of the known categories is still an open problem that has not been resolved [11], making it hard to adapt to task-specific applications like autonomous driving.

On the other hand, limited research has focused on novel object detection for AVs. This is especially crucial because a false-negative detection of unseen objects may result in fatal consequences for AVs. Existing OVOD methods mostly benchmark on datasets of general objects [42, 44] while putting little attention on AV datasets [45–50]. Different from the pursuit of generality in OVOD, perception in AVs has its domain concerns oriented from the image-capturing process by on-car cameras and the object categories due to the scene prior (e.g., road/street objects), which demands

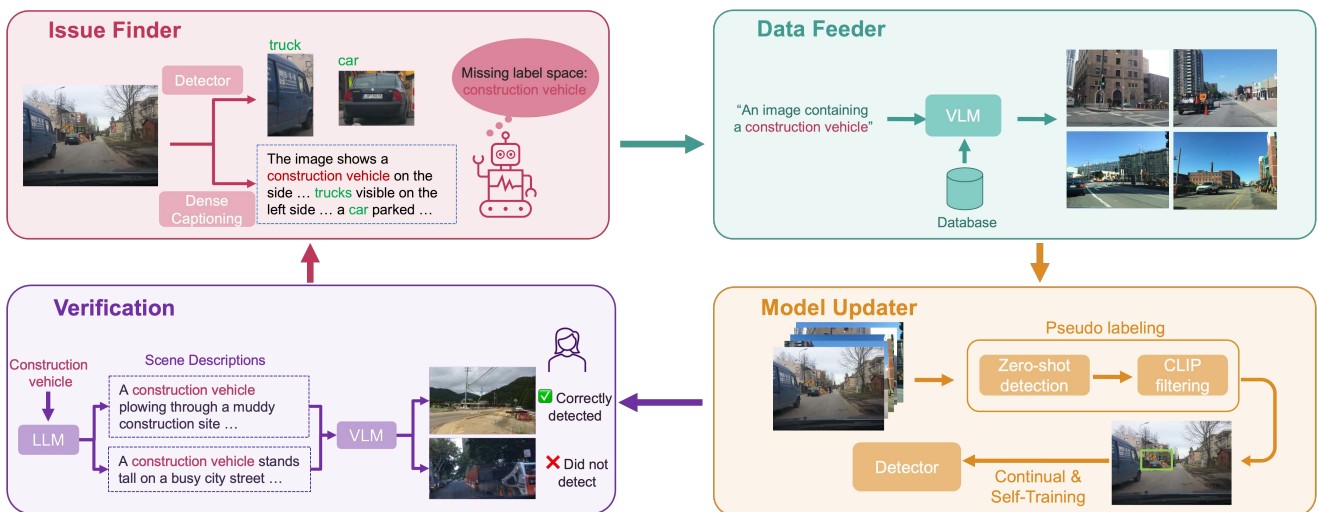

Figure 2. Our design of the automatic data engine includes Issue Finder, Data Feeder, Model Updater, and Verification. The Issue Finder automatically identifies novel categories using the dense captioning model. In the Data Feeder, we employ VLMs to efficiently search for relevant data for training, significantly reducing the inference time for generating pseudo-labels in the subsequent steps and filtering out unrelated images for training. The model is updated in the Model Updater using auto-labeling by VLMs, enabling the recognition of novel categories without incurring any labeling costs. To verify the model, in Verification, we use LLMs to generate descriptions of variations in scenarios and then assess predictions on images queried by VLMs.

task-specific design to enable efficient and scalable system to iteratively enhance AVs on detecting novel objects during its lifecycle. To strike a better trade-off between specificity and generality, our proposed AIDE iteratively extends the closed-set detector's label space so that we can retain decent performance on both novel and known categories for better detection.

**Semi-Supervised Learning (Semi-SL) and Active Learning (AL)** As AVs keep collecting data in operation, a native solution to enable novel category detection is to manually identify the novel category over a collected unlabeled data pool, label them, and then train the detector. Semi-SL [5, 6, 9, 51–54] and AL [8, 10, 18, 55–58] seem to help as they require only a small amount of labeled data to initialize the training. However, labeling even a small amount of data for novel categories will be challenging and costly when given a vast amount of unlabeled data [8, 56, 59–61] by AVs. Moreover, both Semi-SL and AL assume that the labeled and unlabeled data come from the same distribution [51, 62, 63] and share the same label space. However, this assumption does not hold when new categories emerge, inevitably leading to changes in the label space. Naive fine-tuning of the detector only on the novel categories will lead to catastrophic forgetting [64–66] of known categories learned previously. However, Semi-SL methods for object detection do not consider continual learning, while existing continual semi-supervised learning methods [67–70] are also specific to image classification, which is not applicable for object detection.

## 3. Method

This section demonstrates our proposed AIDE, composed of four components: Issue Finder, Data Feeder, Model Updater, and Verification. The Issue Finder automatically identifies missing categories in the existing label space by comparing detection results and dense captions given an image. This triggers the Data Feeder to perform text-guided retrieval for relevant images from the large-scale image pool collected by AVs. The Model Updater then automatically labels queried images and continuously trains the novel category with pseudo-labels on the existing detector. The updated detector is then passed to the Verification module to evaluate under different scenarios and trigger a new iteration if needed. We outline our systematic design in Fig. 2.

### 3.1. Issue Finder

Given the large amount of unlabeled data collected by AVs in daily operation, identifying the missing category of existing label space is difficult as it requires humans to extensively compare the detection results and image context to spot the difference, which hinders the AV system's iterative development. To ease the difficulty, we consider the multi-modality dense captioning (MMDC) models to automate the process. As the MMDC models like Otter [20] are trained with several million multi-modal in-context instruction tuning datasets, they can provide fine-grained and comprehensive descriptions of the scene context as shown in Fig. 3, and we conjecture that they may be more likely to return a synonym to the sought label of the novel category

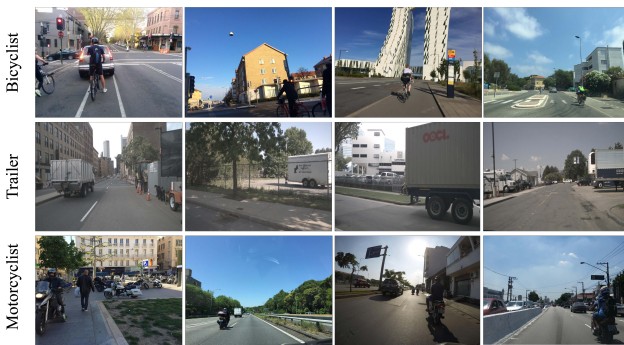

Figure 3. Examples of the Issue Finder. We use Otter [20] to generate detailed descriptions of an image, then identify the novel category that is missing in the label space (shown in red).

Figure 4. Visualization of the queried images from Data Feeder on three novel categories.

than an OVOD method to detect a bounding box for the novel category. Specifically, an unlabeled image will pass to both the detector deployed on-car and the MMDC model to get the list of predicted categories and the detailed captions of the image, respectively. By basic text processing, we can readily identify the novel category the model can not detect. In that case, our data engine will trigger the Data Feeder to query relevant images for incrementally training the detector to extend its label space correspondingly.

## 3.2. Data Feeder

The purpose of Data Feeder is to first query meaningful images that could contain the novel category. The goal is to (1) reduce the search space for pseudo-labeling and accelerate pseudo-labeling in Model Updater, and (2) remove trivial or unrelated images during training so we can reduce training time while also improving performance. This is especially important in real-world scenarios where a large amount of data can be collected every day. As novel categories can be arbitrary and open-vocabulary, a naive solution is to search similar images like the input image of Issue Finder by exploiting the feature similarity, e.g., via similarity of the image feature by CLIP [71]. However, we find that the image similarity cannot reliably identify sufficient numbers of relevant images due to the high variety of the AV datasets (see

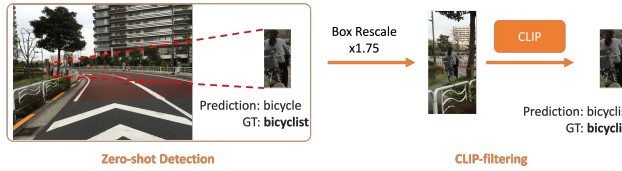

Figure 5. Our two-stage pseudo-labeling for Model Updater: generate boxes by zero-shot detection and label by CLIP filtering.

Tab. 4). Instead, our Data Feeder utilizes the VLMs to perform text-guided image retrieval on the image pool to query for relevant images related to the novel categories. We consider BLIP-2 [21] given its strong open-vocabulary text-guided retrieval capability. Precisely, given an image and a specific text input, we measure the cosine similarity between their embeddings from BLIP-2 and only retrieve the top-$k$ images for further labeling in our Model Updater. For the text prompt, we experiment with common prompt engineering practice [71] and find that a template like *"An image containing {}"* can readily provide good precision and recall for the novel categories in practice. Fig. 4 shows some examples of retrieved images.

## 3.3. Model Updater

The goal of our Model Updater is to make our detector learn to detect novel objects without human annotations. To this end, we perform pseudo-labeling on the images queried by the Data Feeder and then use them to train our detector.

### 3.3.1 Two-Stage Pseudo-Labeling

Motivated by the previous success in pseudo-labeling for object detection [41], we designed our pseudo-labeling procedure with two parts: box and label generation. Such a two-stage framework can help us better dissect the issue of pseudo-label generation and improve the label generation quality. Box generation aims to identify as many object proposals in the image as possible, i.e., high recall for localizing novel categories, to guarantee a sufficient number of candidates for label generation. To this end, region proposal networks (RPN) pretrained with closed-set label space [41] and the open vocabulary detectors (OVD) [11] can be considered, where the former can localize generic objects while the latter can perform text-guided localization. We observe that the SOTA OVD, i.e., OWL-v2 [11] that has been self-trained on web-scale datasets [43], exhibits a higher recall to localize novel categories compared to the RPN. We conjecture that proposals of RPN may be readily biased toward the pre-trained categories.

Thus, we choose OWL-v2 as our zero-shot detector to get the box proposal. Specifically, we append the novel category name provided by Issue Finder to our existing label space and create the text prompts, then we prompt the

OWL-v2 to inference on an image. Note that we only retain the box proposals and remove the labels from the OWL-v2's predictions. This is because we empirically find that OWL-v2 can not achieve reliable precision on the novel categories presented in AV datasets, e.g., less than 10% AP averaging over the novel categories in AV datasets [45, 50], while it can get >40% AP on novel categories of LVIS [42] datasets. We conjecture that this performance degradation may come from the domain shift of the images collected in the AV scenario. For instance, the pretraining data of OWL-v2 mainly comes from the daily image captured by humans from a close distance. However, the street objects are always small in the image due to their long distance from the on-car camera, and the aspect ratio of the image presented in AV datasets is relatively large, making OWL-v2 hard to classify the correct label of the object proposals.

Motivated by this insight, we consider conducting another round of label filtering with CLIP [71] to purify the predictions of the OWL-v2 and generate the pseudo labels. Specifically, we pass the box prediction by OWL-v2 to the original CLIP model [71] for zero-shot classification (ZSC), as shown in Fig. 5. To mitigate the potential issue of the aspect ratio mentioned above, we increase the box size to crop the image and then send the cropped image patch to CLIP for ZSC. This can involve more scene contextual information to help the CLIP better differentiate between novel and known categories. Regarding the label space for CLIP to do zero-shot classification, we first create a base label space, which is a combination of the label space from datasets we have pre-trained and COCO [44], to ensure that we can mostly cover daily objects that would probably be present in the street. The base label space will automatically extend when the Issue Finder identifies novel categories not in the base label space.

### 3.3.2 Continual Training with Pseudo-labels

Directly training our existing detector on the pseudo-labels of novel categories presents a challenge, as these labels may lead the detector to overfit and catastrophically forget the known categories. The issue arises because the unlabeled data can contain both novel and known categories that the detector has previously learned. Without labels for those known categories and only having labels for novel categories, the model may incorrectly suppress predictions for known categories, focusing solely on predicting novel categories. As training progresses, the known categories gradually fade from memory. To address this issue, we draw inspiration from existing self-training strategies and include the pseudo-labels of the known categories that have been trained on. Consequently, our existing detector is updated with the pseudo-labels of both novel and known categories. To obtain pseudo-labels for the known categories, we first

Figure 6. Visualization on the Verification. **LLM output**: We use LLM to generate descriptions of the novel category with variations of the scenarios. **Queried image**: For each description, we use VLM to query images from our training data. **Verification**: we let humans review whether the novel category has been detected.

use our detector to infer data before applying OWL-v2 to the data. Empirically, we find that including pseudo-labels for known categories helps the model distinguish between known and novel categories, boosting the performance of novel categories and mitigating the catastrophic forgetting issues associated with known categories. Additionally, acknowledging that pseudo-labels for both known and novel categories may not be perfect, we filter the pseudo-labels. For known categories, we only use pseudo-labels with high predicted confidence from our detector. For novel categories, we have already incorporated CLIP to filter pseudo-labels, as mentioned in Section 3.3.1.

## 3.4. Verification

The Verification step aims to evaluate whether the updated detector can detect the novel categories under different scenarios, to ensure the model can handle unexpected or unseen scenarios. To this end, we prompt the ChatGPT [12] with the name of novel categories to generate diverse scene descriptions. These descriptions contain variations of the scenarios, such as different appearances of the objects, surrounding objects, time of the day, weather conditions, etc. For each scene description, we again use BLIP-2 to query relevant images, which are used to test the model's robustness. To ensure the correctness, we ask humans to review if the predictions for the novel categories are correct. If the predictions are correct, the detector has passed the unit test. Otherwise, we ask humans to provide the ground-truth label, which can be used to further improve the model. Compared to existing solutions that have humans manually ex-

| Method | Algorithm | Cost ($) | | Accuracy (%) | | |
| --- | --- | --- | --- | --- | --- | --- |
| | | Training | Labeling | Novel | Known | Forgetting |
| Fully-Supervised | | 0.3 | 1005.2 | 24.1 | 29.9 | - |
| Open Vocabulary Object Detection | OwL-ViT [4] | 0.9 | 0 | 2.0 | 5.5 | - |
| | OwL-v2 [11] | 0.9 | 0 | 9.7 | 17.9 | - |
| Semi-Supervised Learning | Unbiased Teacher-v1 [5] | 1.1 | 1.0 | 6.3 | 1.2 | -28.7 |
| AIDE (Ours) | w/o Data Feeder | 5.7 | 0 | 10.1 | 26.8 | -3.1 |
| | w/ Data Feeder | 0.6 | 0 | 12.0 | 26.6 | -3.3 |

Table 1. Cost and accuracy for fully-supervised, open-vocabulary object detection, semi-supervised learning, and our data engine (AIDE) to detect one novel category from Mapillary and nuImages. We initialize Semi-SL and ours with the same detector.

| Method ⟶ | | OVOD | Supervised Training | | | | Semi-SL | AIDE (Ours) | |
| --- | --- | --- | --- | --- | --- | --- | --- | --- | --- |
| Algorithm ⟶ | | OWL-v2 [11] | | | | | UTeacher-v1 [5] | w/o Data Feeder | w/ Data Feeder |
| #Labels per Category ⟶ | | 0 | 10 | 20 | 50 | All | 10 | 0 | 0 |
| Mapillary | motorcyclist | 4.0 | 5.9 | 12.4 | 13.7 | 19.6 | 8.3 | 4.0 | 8.4 |
| Mapillary | bicyclist | 0.9 | 8.9 | 10.8 | 12.4 | 22.4 | 3.5 | 7.7 | 11.9 |
| nuImages | construction vehicle | 4.7 | 3.4 | 8.4 | 7.3 | 22.6 | 4.3 | 5.4 | 5.7 |
| nuImages | trailer | 3.6 | 0.3 | 1.3 | 1.9 | 13.6 | 0.4 | 2.2 | 3.7 |
| nuImages | traffic cone | 35.3 | 12.9 | 21.4 | 28.5 | 42.2 | 16.4 | 31.0 | 30.7 |
| Average | | 9.7 | 6.3 | 10.9 | 12.8 | 24.1 | 6.6 | 10.1 | 12.0 |

Table 2. Per-category accuracy (AP %) on novel categories with different methods.

amine the model prediction one by one, our Verification exploits the LLM to facilitate the search for potential failure cases by diverse scene generation, where the search cost can be largely saved, and the cost of verifying a correct detection or even fixing an incorrect one is lower.

## 4. Experiments

### 4.1. Experimental Setting

**Datasets and Novel Categories Selection** In reality, the AV system can hardly train with a single source of data, e.g., AVs may operate in various locations in the world to collect data. To simulate such a nature faithfully, we leverage the existing AV datasets to jointly train our closed-set detector, including Mapillary [50], Cityscapes [47], nuImages [45], BDD100k [49], Waymo [46], and KITTI [48]. We use this pretrained detector as the initialization for the supervised training, Semi-SL, and our AIDE for a fair comparison. There are 46 categories in total after combining the label spaces. To simulate the novel categories and ensure that the selected categories are meaningful and crucial for AV in the street, we choose 5 categories as novel categories: "motorcyclist" and "bicyclist" from Mapillary, "construction vehicle", "trailer", and "traffic cone" from nuImages. The rest 41 categories are set as known. We remove all the annotations for these categories in our joint datasets and also remove the related categories with similar semantic meanings, e.g., "bicyclist" vs "cyclist". We attach more details of the dataset statistics in the supplementary material.

**Methods for Comparison** To our knowledge, there is little work about the systematic design for automatic data engines tailored to the novel object detection for AV systems. Thus, it is hard to identify a comparable counterpart for our AIDE. To this end, we dissect our evaluation into two parts: (1) compare to alternative detection methods and learning paradigms on the performance of novel object detection; (2) ablation study and analysis of each step of the automatic data engine. For (1), as our AIDE can enable the detector to detect novel categories without any labels, we first compare our method with the zero-shot OVOD methods on novel categories' performance. Moreover, to show the efficiency and effectiveness of our AIDE in reducing label cost, we further compare with semi-supervised learning (Semi-SL) and fully supervised learning that trains the detector with different ratios of ground-truth labels. Specifically, we compare our data engine to state-of-the-art (SOTA) OVOD methods like OWL-v2 [11], OWL-ViT [4], and Semi-SL methods like Unbiased Teacher [5, 6].

**Experimental Protocols** We treat each of the five selected classes as novel classes and conduct experiments separately to simulate the scenario that one novel class has been identified at a time by our Issue Finder. For Semi-SL methods, we provide different numbers of ground-truth images for training. Each image could contain one or multiple objects of the novel category. We evaluate all comparison methods on the dataset of the novel category for a fair comparison.

**Evaluation** As our AIDE automates the whole data curation, model training, and verification process for the AV

system, we are interested in how our engine can strike a balance between the cost of searching and labeling images and the performance on novel object detection. We measure the human labeling costs [72] and also the GPU inference costs [73], i.e., the usage of VLMs/LLMs in our AIDE and training the model with pseudo labeled for our AIDE or with ground-truth labels for comparison methods, denoted as 'Labeling + Training Cost' in Fig. 1. The labeling cost for a bounding box is $0.06 [72], and the GPU cost is $1.1 per hour [73]. The cost of ChatGPT is negligible ($< $0.01).
**Experimental Details** Given the real-time requirement for inference, we choose the Fast-RCNN [22] as our detector instead of OVOD methods like OWL-ViT [4] as the FPS for OWL-ViT is only 3. We run our AIDE to iteratively scale up its capability of detecting novel objects. For multi-dataset training, we follow the same recipe from [74]. For each novel category, we train for 3000 iterations with the learning rate of 5e-4, and we use the same hyperparameter for all the comparison methods if they require training. We attach our full experimental details in the supplementary material.

## 4.2. Overall Performance

In this section, we provide the overall performance of novel object detection after running our AIDE for a complete cycle. Our results are shown in Fig. 1 and Tab. 1. Compared to the SOTA OVOD method, OwL-v2 [11], our method outperforms by 2.3%AP on novel categories and 8.7%AP on known categories, showing that our AIDE can benefit from mining the open-vocabulary knowledge from OVOD method. This is due to our simple yet effective continual training strategy described in Section 3.3.2. Moreover, our AIDE suffers much less from catastrophic forgetting compared to Semi-SL methods, since current Semi-SL methods for object detection do not contain continual learning settings. Existing works on continual semi-supervised learning [67, 70] only consider image classification and are not applicable to object detection. Combining our AIDE with and without the Data Feeder makes it apparent that our Data Feeder can sufficiently reduce the inference time cost as the Data Feeder can pre-filter irrelevant images, and the Model Updater only needs to assign pseudo-labels on a small number of relevant images. Tab. 1 shows that pre-filtering leads to better AP on novel categories.

## 4.3. Analysis on AIDE

In the following subsections, we will dissect each part of our AIDE to validate our design choice.

### 4.3.1 Issue Finder

As mentioned in Section 3.1, the main goal of our Issue Finder is to automatically identify categories that do not exist in our label space. To this end, we evaluate the success

| Dataset | Category Name | Dense Captioning Precision (%) | OVOD AP50 (%) |
|---|---|---|---|
| Mapillary | motorcyclist | 83.3 | 9.5 |
| Mapillary | bicyclist | 89.5 | 1.6 |
| nuImages | const. vehicle | 65.6 | 12.9 |
| nuImages | trailer | 24.7 | 7.1 |
| nuImages | traffic cone | 87.9 | 60.3 |
| Average | | 70.2 | 18.3 |

Table 3. Comparing with using OVOD to identify and localize novel categories, Dense Captioning better predicts missing categories more reliably in our Issue Finder.

| Dataset | Category | Image similarity | VLM Retrieval | |
|---|---|---|---|---|
| | | | CLIP | BLIP-2 |
| Mapillary | motorcyclist | 22.6 | 19.0 | 50.4 |
| Mapillary | bicyclist | 17.9 | 28.8 | 50.5 |
| nuImages | const. vehicle | 14.2 | 51.2 | 55.6 |
| nuImages | trailer | 10.5 | 23.3 | 16.5 |
| nuImages | traffic cone | 29.5 | 47.3 | 99.3 |
| Average | | 18.9 | 33.9 | 54.5 |

Table 4. Ablation studies of the Data Feeder. We report accuracy (%) of the top-$1k$ images queried by image similarity search and text-based retrieval with VLM, i.e., CLIP and BLIP-2.

rate of automatically identifying the novel categories. We find that dense captioning models can automatically predict if the image contains the novel categories more precisely, compared to using OVOD methods to identify and localize novel objects when they are given the names of the novel categories, as shown in Tab. 3. Note that the goal here is to only identify the missing categories, hence we choose to use dense captions here and leverage OVOD to help localize the novel object in the later steps.

### 4.3.2 Data Feeder

The goal of the Data Feeder is to curate relevant data from a large pool of images with high precision. We compare several choices, including image similarity search by CLIP feature, and text-guided image retrieval by VLMs, i.e., BLIP-2 and the CLIP. We report the accuracy of top-$k$ queried images over different categories in Tab. 4, showing that image similarity search is inferior to VLMs. This is because the novel categories can have large intra-class variations, and thus only one image may not be representative of finding sufficient amounts of relevant images. Compared with CLIP, our choice of BLIP-2 performs better on average.

### 4.3.3 Model Updater

We ablate the design choices for our box and pseudo-label generation. For box generation, we compare our choice of using box proposals from OWL-v2 with using proposals

| Category | SAM | VL-PLM | w/o CLIP | ex. known | Ours |
|---|---|---|---|---|---|
| motorcyclist | 0.5 | 10.1 | 3.3 | 2.8 | 8.4 |
| bicyclist | 2.8 | 6.5 | 3.2 | 2.1 | 11.9 |
| const. vehicle | 1.4 | 4.3 | 4.0 | 3.5 | 5.7 |
| trailer | 0.4 | 0.4 | 2.0 | 1.1 | 3.7 |
| traffic cone | 14.5 | 10.4 | 30.0 | 30.9 | 30.7 |
| Average AP (%) | 3.9 | 6.3 | 8.5 | 8.1 | 12.0 |

Table 5. Ablation of Model Updater on box generation with SAM and VL-PLM, label generation without CLIP filtering, and continual training excluded pseudo labels of known categories.

| Dataset | Category | Diversity (%) |
|---|---|---|
| Mapillary | motorcyclist | 57.6 |
| Mapillary | bicyclist | 62.2 |
| nuImages | const. vehicle | 77.0 |
| nuImages | trailer | 82.0 |
| nuImages | traffic cone | 70.4 |
| Average | | 69.8 |

Table 6. Our Verification step can indeed find diverse scenarios. The diversity is measured by the number of distinct images among 100 queried images using descriptions generated by ChatGPT.

from VL-PLM [41], which generates box proposals by the region proposal network (RPN) of MaskRCNN [75] pre-trained on COCO. We also compare with using proposals from Segment Anything model (SAM) [16], specifically we use the FastSAM [76] since it is faster in inference while having the same performance as SAM. As shown in the ablation studies in Tab. 5, our choice of using OWL-v2 is the best among using VL-PLM and SAM. We observe that SAM may generate many small objects with no semantic meaning, suppressing the effective amount of pseudo-labels. This is expected as the pre-training of SAM does not use semantic labels. For label generation, we compare with using OWL-v2 prediction directly without filtering by CLIP, i.e., "w/o CLIP", showing that filtering labels with CLIP is necessary. Last, compared with training our detector without pseudo-labels of known category, denoted as "ex. known", we outperform by 3.9% AP on novel categories. Moreover, the AP of known categories without using pseudo-label is only 1.58%, while Ours is 26.6% as shown in Tab. 1. This verifies the effect of using pseudo-labels of known categories as discussed in Sec. 3.3.2.

### 4.3.4 Verification

The goal of the Verification is to evaluate the detector's robustness and to verify the performance under diverse scenarios. Humans only need to examine if the predictions are correct in each scenario which reduces the monitoring cost since the scenarios are diverse and it takes less time to check the predictions than to annotate. To test if the generated sce-

**LLM output:** A foggy morning image capturing a **motorcyclist** with white hamlet on a countryside trail with lush trees in the background.

| **Queried image:** train | val (before 2nd round training) | val (after 2nd round training) |
|---|---|---|
| 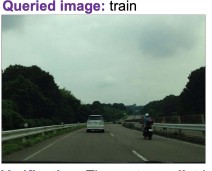 | 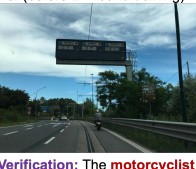 | 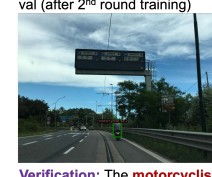 |
| **Verification:** The **motorcyclist** is not detected ✗ | **Verification:** The **motorcyclist** is not detected ✗ | **Verification:** The **motorcyclist** is detected ✅ |

Figure 7. Visualization on the Verification. **Left:** In the queried image from the training set for verification, the model is not predicting the motorcyclist. **Middle:** Similarly on the queried image from the validation set, the model is not predicting the motorcyclist. **Right:** After updating the model again, our model can successfully predict the motorcyclist.

narios are diverse, we measure the number of unique images among 100 images queried by generated descriptions and repeat the process ten times. As shown in Tab. 6, our Verification can indeed find diverse scenarios, as 69.8% images are distinct on average, even on such small training datasets.

If the prediction is incorrect, we can ask annotators to label the images, which are used to further improve the detector. To this end, we randomly select 10 LLM-generated descriptions, for which top-1 retrieved image (based on BLIP-2 cosine similarity) was predicted incorrectly, and labeled these 10 images to update our detector by Model Updater. As shown in Fig. 7, after updating the model with a few human supervisions, our model can successfully predict the object, e.g., the motorcyclist in the figure, which was miss-detected before. For the overall performance, we achieve 14.2% AP on novel categories, which improves our zero-shot performance by 2.2% AP, while the total cost only increases to $1.59. This is still less than $2.1 of semi-supervised learning, and our AP for known categories remains 26.6% after Verification.

## 5. Conclusion

We proposed an Automatic Data Engine (AIDE) that can automatically identify the issues, efficiently curate data, improve the model using auto-labeling, and verify the model through generated diverse scenarios. By leveraging VLMs and LLMs, our pipeline reduces labeling and training costs while achieving better accuracies on novel object detection. The process operates iteratively which allows continuous improvement of the model, which is critical for autonomous driving systems to handle expected events. We also establish a benchmark for open-world detection on AV datasets, demonstrating our method's better performance at a reduced cost. One of the limitations of AIDE is that VLM and LLM can hallucinate in issue finder and verification. Despite the effectiveness of AIDE, for a safety-critical system, some human oversight is always recommended.

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
