# Northwestern

**Mingfu Liang <mingfuliang2020@u.northwestern.edu>**

## [CVPR 2024] Decision notification for your submission 584: AIDE: An Automatic Data Engine for Object Detection in Autonomous Driving

1 message

**OpenReview** <noreply@openreview.net>                                    Mon, Feb 26, 2024 at 5:54 PM
Reply-To: cvpr-2024-pcs@googlegroups.com
To: mingfuliang2020@u.northwestern.edu

Dear Mingfu Liang,

Congratulations! The following paper has been accepted to CVPR 2024:

AIDE: An Automatic Data Engine for Object Detection in Autonomous Driving

Note that acceptance is contingent on the paper passing an iThenticate plagiarism check.

You can access the decision and the final reviewer comments here:
https://openreview.net/forum?id=tXY50pjnSV

We hope that you will be able to use this feedback to improve the camera-ready version of your paper.

We will send more information in the next few days, including whether your paper will be a poster or oral presentation, detailed statistics about acceptance rate, and instructions on how to prepare the camera-ready copy. The camera-ready deadline will be March 25.

We hope to see you at CVPR!

Ali Farhadi, David Crandall, Imari Sato, Jianxin Wu, Robert Pless, Zeynep Akata
CVPR 2024 Program Chairs
and
David Forsyth
Senior Advisor to the CVPR PCs

# AIDE: An Automatic Data Engine for Object Detection in Autonomous Driving

## Supplementary Material

## A. Verification can Boost AIDE's Performance

In Verification, humans are asked to verify the predictions on the diverse scenarios generated by LLMs (ChatGPT [1]). If the prediction is incorrect, annotators can give correct bounding boxes, which can be used by AIDE to self-improve the model. In this section, we examine whether these annotations can boost the performance of AIDE. To this end, we train the model after we have collected annotations for 10, 20, and 30 images. However, since we only have a few human annotations collected, directly combining them with a large number of pseudo-labels from the Model Updater will cause issues if we have a uniform sampling rate on the data loader during training.

On the other hand, semi-supervised learning methods like Unbiased Teacher-v1 [2] have demonstrated notable performance on novel categories with minimal annotations, owing to their strong augmentation strategy.

Motivated by this insight, we first use the few labeled images to train an auxiliary model by the strong augmentation strategy as [2] but with 1000 iterations to reduce training costs. This auxiliary model is then used to generate pseudo-labels for the novel categories based on the images initially queried by our Data Feeder, and these are combined with the earlier pseudo-labels generated by our Model Updater for both novel and known categories to fine-tune our detector again in our Model Updater. By doing so, we can obtain more pseudo-labels for novel categories with high quality and alleviate the sampling issue in the data loader. As shown in Fig. 1, our AIDE can be largely improved.

## B. More Comparisons between AIDE and OVOD (OWL-v2)

In this section, we demonstrate that AIDE is a general automatic data engine that can enhance different object detectors for novel object detection. Specifically, we replace the closed-set detector (Faster RCNN [3]) with the state-of-the-art (SOTA) open-vocabulary object detection (OVOD) method, OWL-v2.

As shown in Tab. 1, by applying our AIDE on OWL-v2, we can achieve 13.2% AP on average without human annotations, marking a 3.5% improvement over the original OWL-v2 model. However, our default detector is Faster RCNN since it has a faster inference speed, which is favorable for autonomous driving.

In addition, the original OWL-v2 paper [4] proposes a self-training strategy to enhance the OWL-v2 on novel object detection, i.e., directly using the predictions of OWL-v2

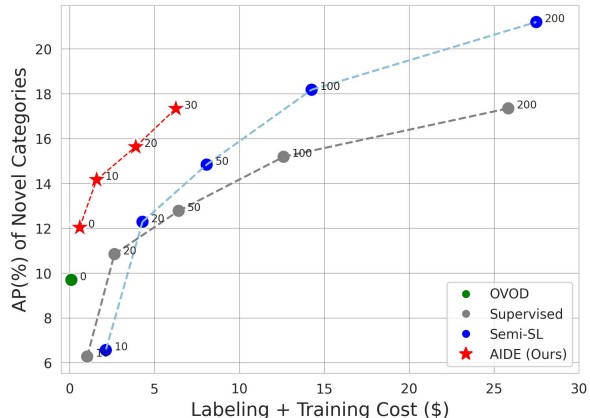

Figure 1. We demonstrate that the annotations in the Verification step can boost the performance of AIDE. The numbers next to the data points denote the number of labeled images used by each method. Note that AIDE only introduces labeled images in Verification if an annotator wants to provide the labels when the detector gives incorrect predictions on the test scenarios.

| Categoty | OVOD | | AIDE (Ours) | |
|---|---|---|---|---|
| | OWL-v2 | OWL-v2 ST | Faster RCNN | OWL-v2 |
| motorcyclist | 4.0 | 5.3 | 8.4 | 11.4 |
| bicyclist | 0.9 | 0.8 | 11.9 | 9.8 |
| const. vehicle | 4.7 | 5.4 | 5.7 | 6.0 |
| trailer | 3.6 | 3.5 | 3.6 | 3.6 |
| traffic cone | 35.3 | 35.5 | 30.7 | 35.3 |
| Average AP(%) | 9.7 | 10.1 | 12.0 | 13.2 |

Table 1. Comparison between OWL-v2, OWL-v2 with self-training, and AIDE on improving an existing detector on novel object detection with any human annotations. ST: Self-training using the same strategy in [4].

with a certain confidence threshold to self-train the OWL-v2. We compare this self-training schedule with our AIDE.

As shown in Tab. 1, the self-training can improve the OWL-v2, but it is still inferior to AIDE 3.1%. This improvement is attributable to our Data Feeder and the CLIP filtering in our Model Updater, which help to minimize irrelevant images for pseudo-labeling and filter out inaccurate OWL-v2 predictions, thereby enhancing the quality of pseudo-labels and the subsequent performance after fine-tuning OWL-v2 with these labels. We will dissect the impact of our Data Feeder and Model Updater on improving the quality of pseudo-label in Sec. D.2 and Tab. 4.

| Dataset | Category | Mapillary / nuImages | +Waymo (39k) | +Waymo (78k) | +Waymo (78k) +BDD100k (69k) |
|---------|----------|----------------------|--------------|--------------|------------------------------|
| Mapillary | motorcyclist | 8.4 | 9.4 | 11.1 | 13.4 |
| Mapillary | bicyclist | 11.9 | 13.0 | 15.0 | 18.4 |
| nuImages | const. vehicle | 5.7 | 7.3 | 14.6 | 19.7 |
| nuImages | trailer | 3.7 | 3.6 | 5.1 | 11.2 |
| nuImages | traffic cone | 30.7 | 31.6 | 35.1 | 36.1 |
| Average AP(%) | | 12.0 | 13.9 | 16.2 | 19.8 |

Table 2. Extending the image pool with the Waymo and BDD100k dataset in Data Feeder can boost the performance of AIDE.

## C. Extending the Image Pool further boosts AIDE's Performance

Our Data Feeder queries images from either Mapillary [5] or nuImages [6] by default. To verify the scalability of AIDE, we add the Waymo dataset in the database for Data Feeder, i.e., the image pool for querying becomes {nuImages, Waymo} or {Mapillary, Waymo} for each novel category. Note that the Waymo dataset only contains three coarse labels, i.e., "vehicle", "pedestrian", and "cyclist", as shown in Tab. 5. Therefore it is uncertain whether novel categories such as "motorcyclist", "construction vehicle", "trailer", and "traffic cone" are present in the Waymo dataset. For "bicyclist", although the Waymo dataset includes a similar label "cyclist", we have excluded all annotations of this category as described in Sec. 4.1 of our main paper. Moreover, given that the Waymo dataset consists largely of videos, resulting in numerous similar images, we implemented a sampling strategy. Each video was subsampled with a frame rate of 20, reducing the total number of images from 790,405 to 39,750 (denoted as 39k). We used the same hyperparameters for BLIP-2 and CLIP in our Data Feeder and Model Updater as were used for the Mapillary and nuImages datasets, respectively, for image querying and pseudo-labeling.

As indicated in Table 2, incorporating the Waymo dataset into our Data Feeder for image querying resulted in a 1.9% AP improvement in detecting novel categories, compared to using only the Mapillary or nuImages datasets. Moreover, by adding more unlabeled images from Waymo and the full BDD100k dataset, we can boost the performance to 19.8% AP, approaching the fully-superivsed result of 24.1% AP. Note that the cost of AIDE is only $2.4 with 19.8% AP. This significant improvement demonstrates that our AIDE can effectively scale up with an expanded image search space.

## D. More Analysis

### D.1. Ablation Study of the Scaling Ratio for CLIP filtering

As discussed and illustrated in Sec. 3.3.1 and Fig. 5 of our main paper, we increase the size of the pseudo-box used to

| Dataset | Category Name | Scaling Ratio | | | | |
|---------|---------------|---|------|-----|------|---|
| | | 1 | 1.25 | 1.5 | 1.75 | 2 |
| Mapillary | motorcyclist | 3.6 | 6.1 | 7.6 | 8.4 | 8.9 |
| Mapillary | bicyclist | 9.3 | 10.7 | 12.0 | 11.9 | 12.2 |
| nuImages | cons. vehicle | 5.8 | 5.0 | 4.8 | 5.7 | 5.4 |
| nuImages | trailer | 2.1 | 2.1 | 3.2 | 3.6 | 3.6 |
| nuImages | traffic cone | 28.6 | 30.2 | 28.6 | 30.7 | 29.2 |
| Average AP(%) | | 9.9 | 10.8 | 11.2 | **12.0** | 11.8 |

Table 3. Ablation study of the scaling ratio of the pseudo-box to crop the image patch for CLIP filtering.

crop the image before submitting the cropped image patch for zero-shot classification (ZSC). We present an ablation study of the scaling ratio, ranging from 1.0 to 2.0, where a scaling ratio of 1.0 signifies using the pseudo-box dimensions as they are to crop the image patch. As Table 3 demonstrates, the performance of novel categories improves as the scaling ratio increases, reaching a plateau when the scaling ratio is 1.75. This trend is expected since a substantially rescaled box might include excessive background context, potentially distracting the ZSC process of CLIP. Therefore, we use a scaling ratio of 1.75 for all our experiments.

### D.2. Analyzing the Data Feeder and Model Updater on Improving the Quality of Pseudo-labeling

We analyze the impact of our Data Feeder and Model Updater on improving the quality of pseudo-labels. As outlined in Section 3.2 of our main paper, our Data Feeder is designed to query images relevant to novel categories from the image pool. This process helps eliminate trivial or unrelated images during training, thereby reducing training time and enhancing performance. Moreover, our two-stage pseudo-labeling in our Model Updater will filter out raw pseudo-labels generated by OWL-v2.

To establish a baseline for comparison, we initially used OWL-v2 to perform inference on the entire image pool, i.e., Mapillary or nuImages datasets for each novel category. We measured the precision of the pseudo-labels for novel categories against the ground-truth labels in each dataset, considering a pseudo-label as a true positive if it achieved an Intersection over Union (IoU) greater than 0.5 with the

| Category | OWL-v2 [4] | w/ Data Feeder | w/ Model Updater |
|---|---|---|---|
| motorcyclist | 11.1 | 19.3 | 47.2 |
| bicyclist | 5.3 | 7.6 | 33.8 |
| const. vehicle | 11.3 | 12.8 | 16.5 |
| trailer | 10.9 | 12.1 | 38.2 |
| traffic cone | 68.3 | 76.9 | 92.9 |
| Average AP(%) | 21.4 | 25.7 | 45.7 |

Table 4. Evaluate the quality of the pseudo-labels of novel categories generated by OWL-v2 without any post-processing, filtered by the Data Feeder with BLIP-2, and further filtered by Model Updater. We measure the precision (%) by comparing the pseudo labels with ground-truth labels for each novel category. Given a pseudo-label, we treat it as a true positive if it has an IoU larger than 0.5 with the ground-truth label, otherwise it is a false positive.

ground truth. This baseline performance sets the stage for appreciating the enhancements brought by our Data Feeder and Model Updater. Following this, we report on the precision of pseudo-labels after image-level filtering by our Data Feeder and pseudo-label filtering by our Model Updater.

Table 4 shows that compared to the raw pseudo-labels generated by OWL-v2, our Data Feeder alone improved the average precision of novel categories by 4.3%. Furthermore, when combined with our Model Updater, the average precision was enhanced to 45.7%, which is a 24.3% improvement over the raw pseudo-labels from OWL-v2. This significant improvement underscores the effectiveness of our AIDE in fine-tuning OWL-v2, surpassing the self-training method proposed by OWL-v2 in [4], as our AIDE provides substantially better quality pseudo-labels.

## E. Limitations

Our work proposed the first automated data engine, AIDE, based on VLMs and LLMs for autonomous driving. However, there are still limitations in our work. As AIDE is extensively integrated with VLMs and LLMs, the hallucination of VLMs and LLMs may have negative impacts on our Issue Finder and Verification. Although the dense captioning model in our Issue Finder can automatically identify the novel category with high precision, it may also potentially hallucinate novel categories that are not present in the image. On the other hand, although our Verification can generate diverse scene descriptions for evaluating our detector, it may also hallucinate scenarios that do not exist in the image pool.

Generally, we believe that these concerns will be alleviated with the advancement of VLMs and LLMs in the future. Additionally, using a large image pool for text-based retrieval in Data Feeder can help mitigate these concerns. Despite the effectiveness of AIDE, for a safety-critical system, some human oversight is always recommended.

## F. More Experimental Details

In this section, we provide more experimental details for our AIDE and also the comparison methods. For all approaches, including supervised training, semi-supervised learning, and AIDE, we begin with the same Faster RCNN model pretrained by the same six AV datasets then proceed to conduct our experiments. For the Unbiased Teacher-v1 [2], we use the official implementation[1] and adhere to the same training settings. Both Supervised Training and AIDE are trained for 3000 iterations, using SGD optimization with a batch size of 4, a learning rate of 5e-4, and weight decay set at 1e-4 across all experiments. The Unbiased Teacher-v1 [2] requires a warm-up stage to pre-train a teacher model, so we allocate an additional 1000 iterations, totaling 4000 iterations, for training this method. All other training hyperparameters for the Unbiased Teacher-v1 [2] remain consistent with those used for Supervised Training and AIDE. For the image-text matching in Data Feeder, we leverage the 'pretrain' configuration to initialize the BLIP-2 model, which is exactly based on the official BLIP-2 GitHub Repo[2]. The VLMs we used are allowed for commercial usage (i.e., Otter/CLIP/BLIP-2). ChatGPT can be replaced by open-source LLMs like Llama2 [7], whereas the cost of ChatGPT is negligible (less than $0.01).

### F.1. Model Hyperparameters for Data Feeder and Model Updater

In this section, we detail the model hyperparameter selection for our Data Feeder and Model Updater. Within our Data Feeder, we utilize BLIP-2 to query images relevant to each novel category. This is achieved by measuring the cosine similarity score between the text and image embeddings. Subsequently, all images are ranked based on their cosine similarity score (denoted as the BLIP-2 score), and the top-ranked images are selected by thresholding the BLIP-2 score. We have set the BLIP-2 score threshold at 0.6 for all novel categories. This threshold is chosen to ensure that our Data Feeder retrieves at least 1% of the images from the image pool (comprising either Mapillary or nuImages datasets) for each novel category. Such a threshold guarantees that we have a sufficient number of images for pseudo-labeling in Model Updater.

Second, in our Model Updater, given that the number of relevant images has been significantly reduced following the BLIP-2 querying process (for example, only 550 images for "motorcyclist"), we opt for a CLIP score threshold, specifically 0.1, for our two-stage pseudo-labeling to prevent excessive filtering out of too many potential pseudo-labels. As demonstrated in Section D.2 and Table 4, even

---

[1] https://github.com/facebookresearch/unbiased-teacher
[2] https://github.com/salesforce/LAVIS/blob/main/examples/blip2_image_text_matching.ipynb

with such a CLIP score threshold, we can still markedly enhance the quality of pseudo-labels compared to using only the Data Feeder to filter OWL-v2's pseudo-labels. For filtering pseudo-labels of known categories, we set the confidence score threshold at 0.6. This threshold significantly reduces the number of pseudo-labels for each known category, helping to balance it with the number of pseudo-labels for novel categories. Such a balance is crucial in mitigating forgetting while simultaneously boosting performance for novel categories.

## F.2. Experimental Details for fine-tuning OWL-v2 with AIDE

For the experiment of fine-tuning the OWL-v2 [4] with AIDE, we leverage the official model released by the author [3]. We opted to use the Hugging Face Transformers library to fine-tune the OWL-v2 [4] as it provides a consistent codebase for both inferring and training OWL-v2 in PyTorch. Notably, the OWL-v2 [4] was self-training on the OWL-ViT [8] on a web-scale dataset, i.e., WebLI [9], and the fine-tuning learning rate is 2e-6. To enable effective continual fine-tuning with AIDE, we set the initial learning rate as 1e-7. This setting is intended to prevent dramatic changes in the weights of OWL-v2, thereby avoiding catastrophic forgetting while still allowing the model to learn novel categories using AIDE effectively. We utilize the same training hyperparameters from the self-training recipe of OWL-v2 [4] to conduct self-training of OWL-v2 on AV datasets in Section B, ensuring a fair comparison.

## F.3. Details for our Verification

As mentioned in our main paper Sec. 3.4, we leverage LLM, i.e., ChatGPT [1], to generate diverse scene descriptions to evaluate the updated detector from our Model Updater. The prompt template we use for this purpose is illustrated in Figure 2. Further, we have detailed the training process triggered by Verification in Section B. We use the same training and model hyperparameters for our continual training in Model Updater when conducting the training triggered by Verification.

## G. More Visualizations

### G.1. Predictions with Different Methods

We present additional visualization results in Figures 3, 4, and 5. These visualizations reveal that the Semi-Supervised Learning (Semi-SL) method tends to overfit to novel categories, resulting in numerous false positive predictions. Furthermore, the Semi-SL method struggles to

---

[3] https://github.com/google-research/scenic/tree/main/scenic/projects/owl_vit
[4] https://huggingface.co/docs/transformers/model_doc/owlv2

---

```
role: "system",
content: "You are a helpful assistant."

role: "user",
content: "Generate variations of descriptions on the appearance of this one to search for autonomous driving data by CLIP:

`An image containing a {novel category}`

Emphasize scenarios with:
(1) the color of the object,
(2) appearances of the object,
(3) scenarios,
(4) type of the road,
(5) surrounding objects,
(6) vehicle density,
(7) different time of the day,
(8) weather,
(9) position of the object, etc.

Return only a list of descriptions in a list without other text."
```

Figure 2. Prompt template for ChatGPT to generate diverse testing scenarios in Verification. The "novel category" is a placeholder in the template and will be replaced by the exact name of the novel category obtained in Issue Finder.

detect known categories, indicating an issue with catastrophic forgetting. In contrast, the state-of-the-art Open-Vocabulary Object Detection (OVOD) method, specifically OWL-v2, also produces many false positives for both novel and known categories. However, compared to both the Semi-SL and OVOD methods, AIDE demonstrates superior performance in accurately detecting both novel and known categories.

## G.2. Prediction after updating our model by Verification

In Figure 6, we present additional visualizations to Fig. 7 in our main paper to demonstrate that an extra round of training, initiated by Verification, further reduces both missed and incorrect detections of novel categories. These visualizations illustrate the effectiveness of the additional training round in enhancing the accuracy and reliability of our detection system for these novel categories.

## H. Discuss about de-duplication process for video data

The nuImages dataset contains 13 frames per scene, spaced 0.5 seconds apart. Currently, we directly use all unlabeled images of nuImages dataset for Data Feeder to query without using any de-duplication process in our main paper. In practice, as the dataset gets larger or with a higher frame rate, de-duplication could further improve the data diversity for querying in Data Feeder and may potentially improve the performance of AIDE, and we leave this for future study.

# I. Comparison between Verification and Active Learning alternatives

We compare our approach, "LLM description+BLIP-2" for Verification, with two Active Learning (AL) baselines. The first one is to verify the boxes predicted as the novel target class by the detector but with the highest classification entropy. The second one is to perform verification on randomly sampled boxes predicted as the novel target class by the detector. For both AL baselines, we use them to verify 10 images, the same as what we have done in Sec. 4.3.4 of our main paper. The two AL baselines only achieve 13.1% and 12.7% AP on novel classes, respectively. This is inferior to our approach (14.2% AP) which uses VLM/LLM to identify diverse AV scenarios for verification.

# J. Discussion for the real-cost of supervised and semi-supervised methods

In our main paper Fig. 1, Tab. 1, and Tab. 2, we only measure the "Labeling and Training" cost for the supervised/semi-supervised methods. In fact, the real cost for the supervised/semi-supervised method is not just labeling images but also includes *searching over the large data pool to find relevant images* to label. For instance, an annotator needs to examine 874 images on average to find 50 images for a selected novel class, costing $43.7 for supervised/semi-supervised methods, assuming it costs 10 seconds per image to inspect for novel classes, which corresponds to $0.05 at $18 per hour. Therefore, AIDE is more practical than supervised/semi-supervised methods for car companies as we automate data querying in Data Feeder to largely reduce the total cost.

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

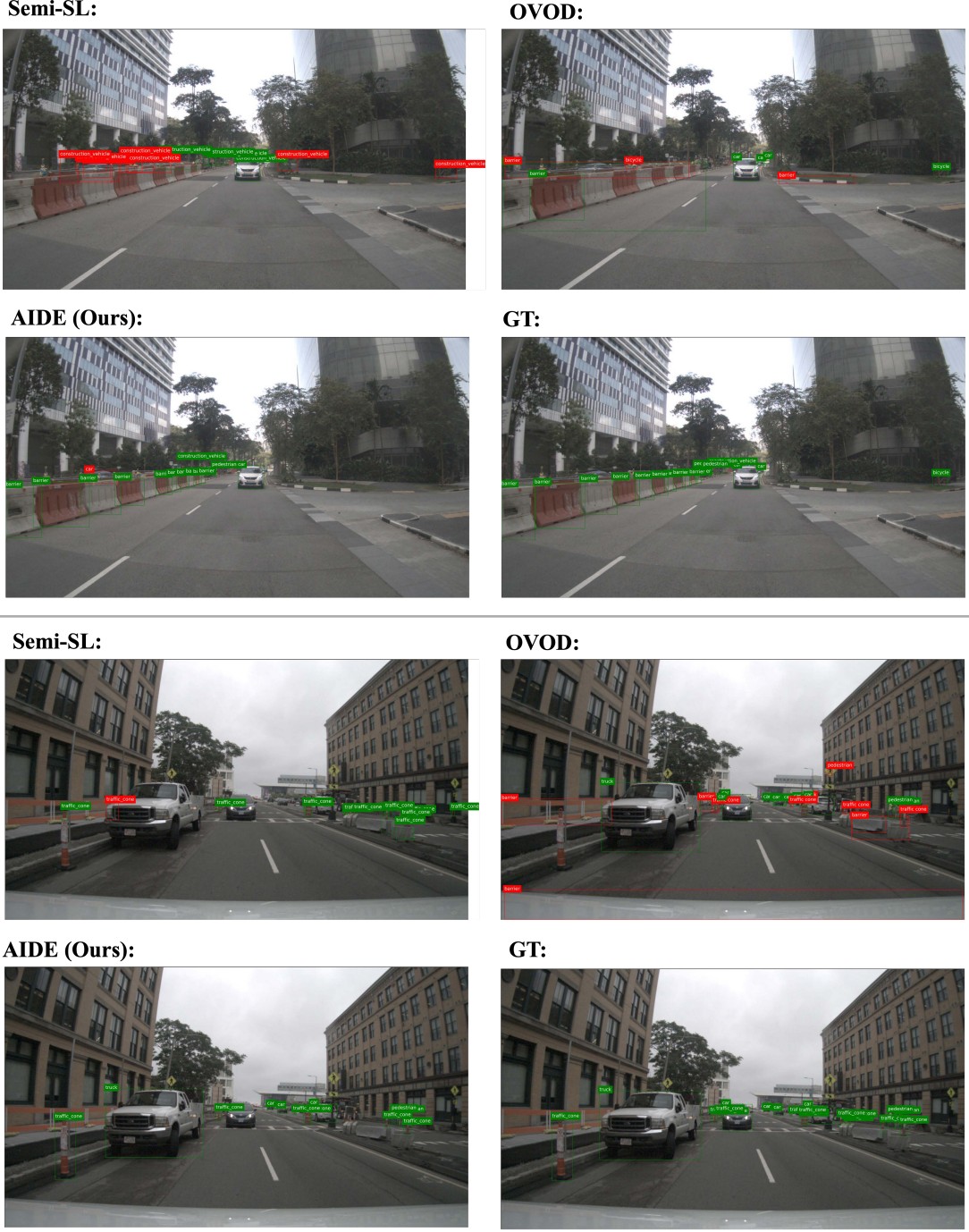

Figure 3. Visualization of the detection results under different methods. We treat a box prediction as true positive if it has an IoU larger than 0.5 with the ground-truth box. The true positive predictions are in green color, while the false positive predictions are in red color. **Top-left**: Semi-supervised Learning (Semi-SL) method, i.e., Unbiased Teacher-v1 [2]. **Top-right**: Open-vocabulary object detection (OVOD) method, i.e., OWL-v2 [4]. **Bottom-left**: AIDE. Bottom-right: ground-truth.

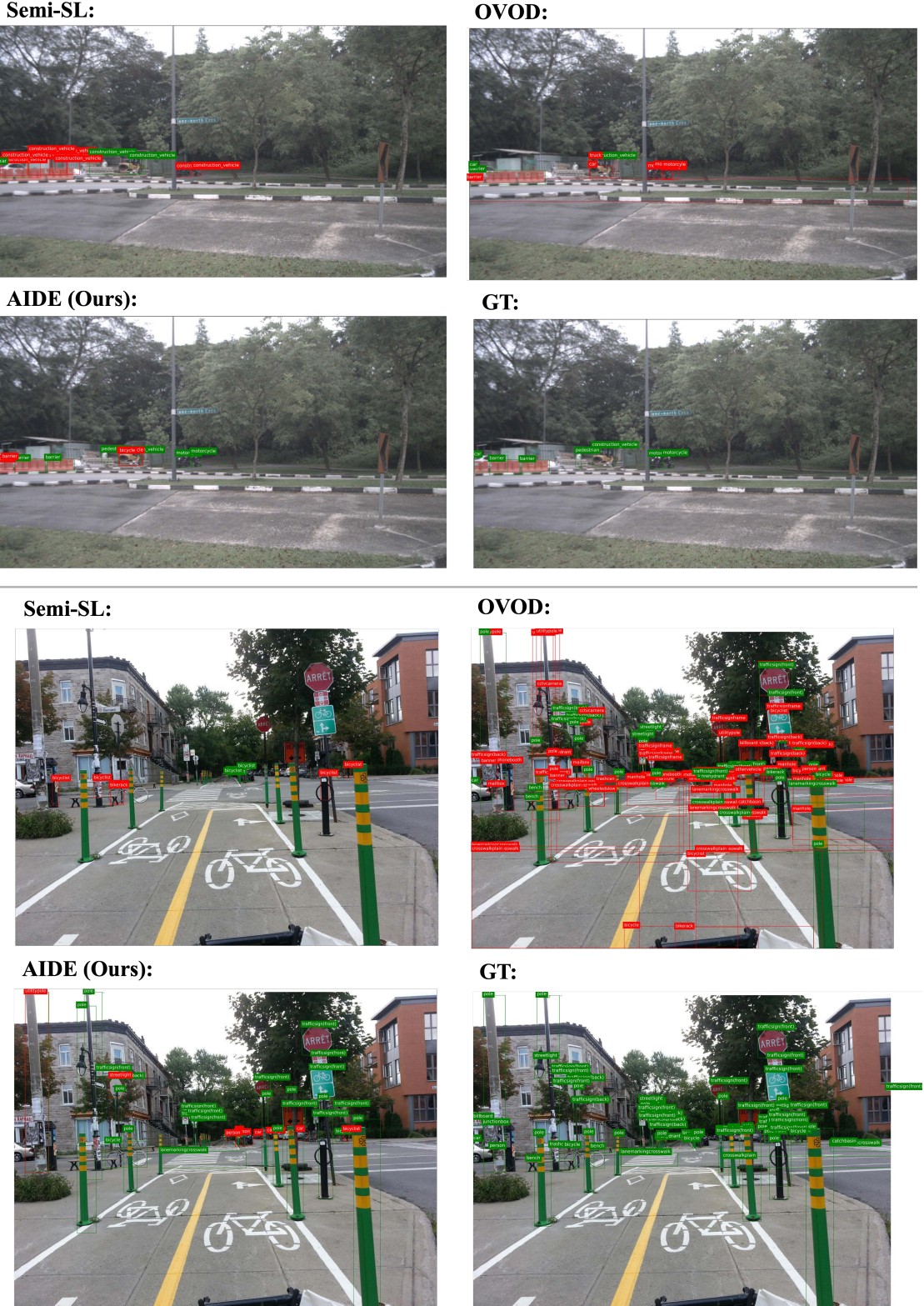

Figure 4. Visualization of the detection results under different methods. We treat a box prediction as true positive if it has an IoU larger than 0.5 with the ground-truth box. The true positive predictions are in green color, while the false positive predictions are in red color. **Top-left**: Semi-supervised Learning (Semi-SL) method, i.e., Unbiased Teacher-v1 [2]. **Top-right**: Open-vocabulary object detection (OVOD) method, i.e., OWL-v2 [4]. **Bottom-left**: AIDE. **Bottom-right**: ground-truth.

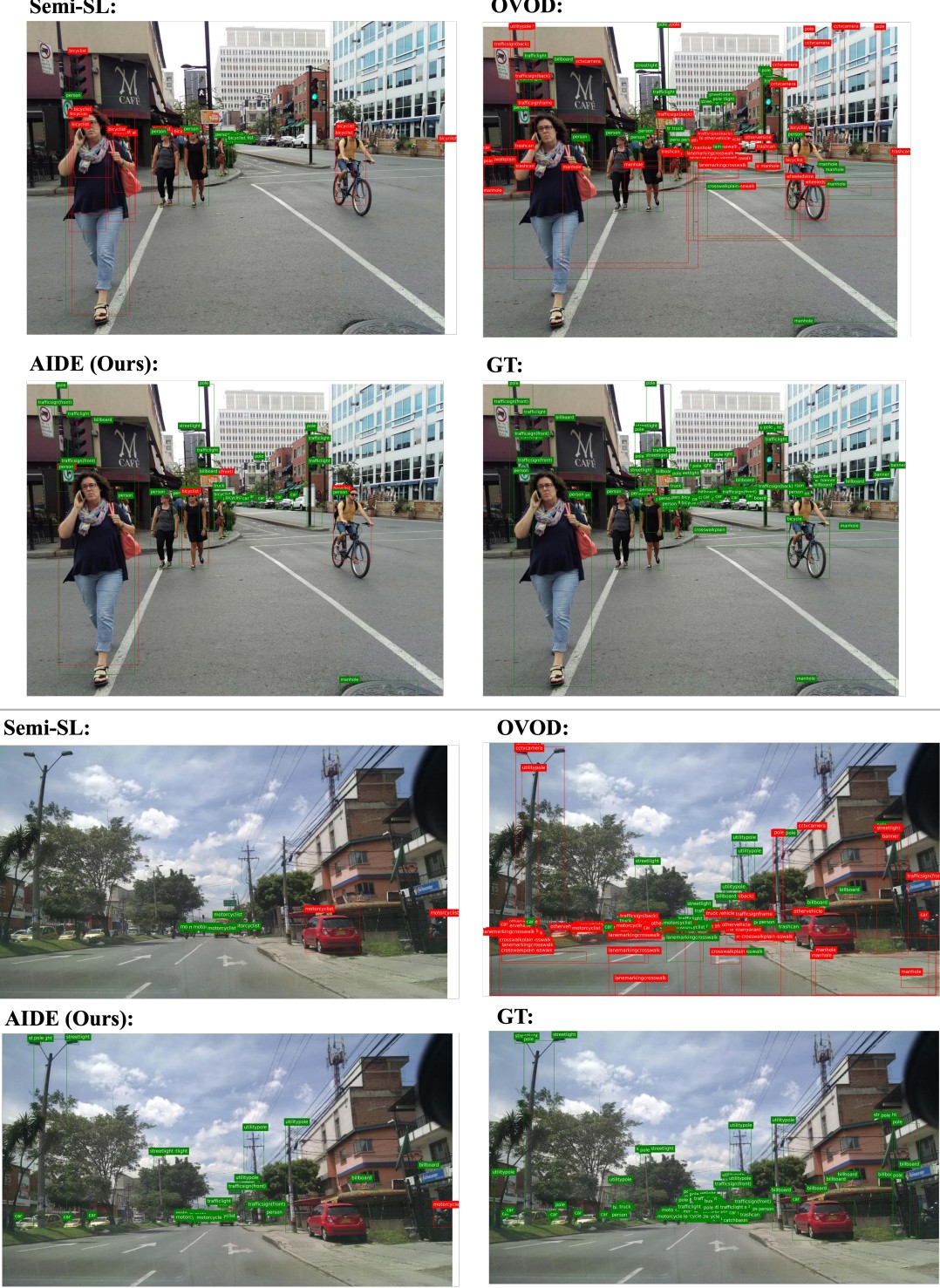

Figure 5. Visualization of the detection results under different methods. We treat a box prediction as true positive if it has an IoU larger than 0.5 with the ground-truth box. The true positive predictions are in green color, while the false positive predictions are in red color. **Top-left**: Semi-supervised Learning (Semi-SL) method, i.e., Unbiased Teacher-v1 [2]. **Top-right**: Open-vocabulary object detection (OVOD) method, i.e., OWL-v2 [4]. **Bottom-left**: AIDE. **Bottom-right**: ground-truth. Note that some original annotations in Mapillary are not correct. For instance, for the image of "GT" in the second row, the human on the bicycle should be labeled as "bicyclist" while the original label is "person".

**LLM output:** A daytime image depicting a vibrant red **motorcyclist** moving on a busy city road.

**Queried image:** train

val (before 2nd round training)

val (after 2nd round training)

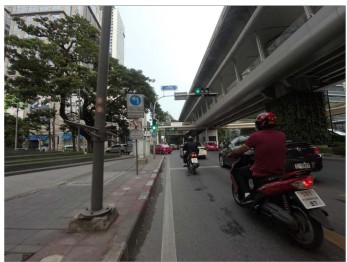 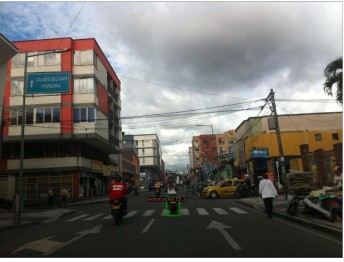 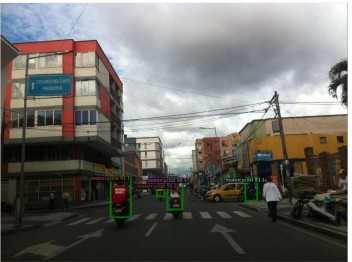

**Verification:** The red **motorcyclist** is not detected ❌

**Verification:** The red **motorcyclist** is not detected ❌

**Verification:** The red **motorcyclist** is detected ✅

**LLM output:** A dusk scene showing a black motorcyclist wearing a black helmet, maneuvering through heavy traffic on a crowded urban street.

**Queried image:** train

val (before 2nd round training)

val (after 2nd round training)

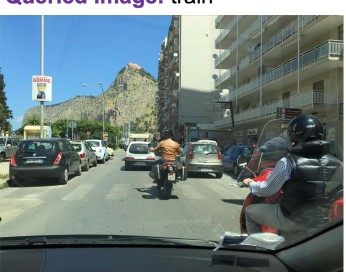 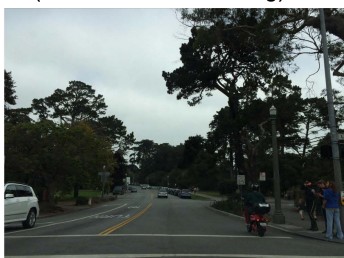 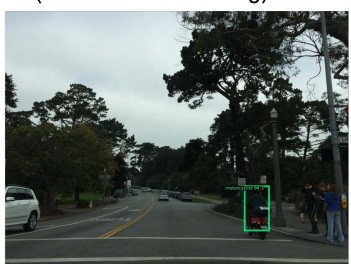

**Verification:** The **motorcyclist** is not detected ❌

**Verification:** The **motorcyclist** is not detected ❌

**Verification:** The **motorcyclist** is detected ✅

Figure 6. More visualizations on our Verification. **Left:** In the queried image from the training set for verification, the model is not predicting the motorcyclist. **Middle:** Similarly on the queried image from the validation set, the model is not predicting the motorcyclist. **Right:** After updating the model again, our model can successfully predict the motorcyclist.

|  | Cityscapes | KITTI | BDD100k | nuImages | Mapillary | Waymo |
|---|---|---|---|---|---|---|
| # Classes | 8 | 3 | 10 | 10 | 37 | 3 |
| Cumulative # Classes | 8 | 10 | 12 | 16 | 45 | 46 |
| # Images | 2,975 | 6,859 | 69,863 | 67,279 | 18,000 | 790,405 |
| Vehicle | car truck bus train motorcycle bicycle | car | car truck bus train motorcycle bicycle | car truck bus motorcycle bicycle construction vehicle trailer | car truck bus motorcycle bicycle trailer caravan boat wheeled-slow other vehicle | vehicle |
| Human | person rider | pedestrian cyclist | pedestrian rider | pedestrian motorcyclist | person bicyclist other rider | pedestrian cyclist |
| Traffic Objects |  |  | traffic light traffic sign | traffic cone barrier traffic sign frame | traffic light traffic sign(back) traffic sign(front) pole street light utility pole |  |
| Other Objects |  |  |  |  | bird ground animal crosswalk plain lane marking crosswalk banner bench bike rack billboard catch basin cctv camera fire hydrant junction box mailbox manhole phone booth trash can |  |

Table 5. The statistics and label space of the six AV datasets, i.e., Cityscapes [10], KITTI [11], BDD100k [12], nuImages [6], Mapillary [5], and Waymo [13]. There are 46 categories in total after combining the label spaces. To simulate the novel categories and ensure that the selected categories are meaningful and crucial for AV in the street, we choose 5 categories as novel categories: "motorcyclist" and "bicyclist" from Mapillary, "construction vehicle", "trailer", and "traffic cone" from nuImages. The rest 41 categories are set as known. We remove all the annotations for these categories in our joint datasets and also remove the related categories with similar semantic meanings, e.g., "bicyclist" vs "cyclist", "rider" vs "motorcyclist".