# OpenReview forum: "AIDE: An Automatic Data Engine for Object Detection in Autonomous Driving"
_thecvf.com/CVPR/2024/Workshop/VLADR — VLADR 2024 Oral_

### Official Review · Reviewer_fZ2p · 2024-04-17

**Rating:** 7
**Confidence:** 5

**Review:**

This paper proposes an impressive and comprehensive Automatic Data Engine (AIDE) that leverages vision-language models (VLMs) and large language models (LLMs) to automate the entire data engine pipeline for open-world object detection in autonomous driving. The authors have put together a thoughtful and well-designed system that addresses the key challenges in this domain.

Strengths:

The authors have identified a crucial problem in autonomous driving - the need for continuous model improvement to handle the long-tailed distribution of objects encountered on the road. Their proposed AIDE system addresses this in a systematic and automated way.
The use of VLMs and LLMs to power the various components of the data engine (Issue Finder, Data Feeder, Model Updater, Verification) is a clever and effective application of these powerful models.
The detailed experiments and analysis demonstrate the effectiveness of AIDE, showing significant improvements over baselines like open-vocabulary object detection and semi-supervised learning, while reducing labeling costs.
The proposed benchmark for open-world detection on AV datasets is a valuable contribution that can enable further research in this direction.

Weaknesses:

The paper could benefit from a more thorough discussion of the limitations and potential failure modes of the AIDE system, especially around the reliability of the VLM and LLM components.
While the authors mention the need for some human oversight for a safety-critical system like autonomous driving, more details on the human-in-the-loop aspects could strengthen the paper.
The connection between the individual components of AIDE and how they work together cohesively could be explained in a more intuitive way.

Recommendation: Overall, this is a strong and well-executed paper that makes a significant contribution to the field of autonomous driving. I recommend acceptance, subject to addressing the minor points raised above in the camera-ready version.

---

### Decision · Program_Chairs · 2024-04-22

Accept (Oral)